# Diversity and N_2_O Production Potential of Fungi in an Oceanic Oxygen Minimum Zone

**DOI:** 10.3390/jof7030218

**Published:** 2021-03-17

**Authors:** Xuefeng Peng, David L. Valentine

**Affiliations:** 1Marine Science Institute, University of California, Santa Barbara, CA 93106, USA; valentine@ucsb.edu; 2School of Earth, Ocean and Environment, University of South Carolina, Columbia, SC 29208, USA; 3Department of Earth Science, University of California, Santa Barbara, CA 93106, USA

**Keywords:** marine fungi, oxygen minimum zone, nitrous oxide, diversity, ^15^N tracer, size-fractioned, eastern tropical North Pacific, metagenome

## Abstract

Fungi in terrestrial environments are known to play a key role in carbon and nitrogen biogeochemistry and exhibit high diversity. In contrast, the diversity and function of fungi in the ocean has remained underexplored and largely neglected. In the eastern tropical North Pacific oxygen minimum zone, we examined the fungal diversity by sequencing the internal transcribed spacer region 2 (ITS2) and mining a metagenome dataset collected from the same region. Additionally, we coupled ^15^N-tracer experiments with a selective inhibition method to determine the potential contribution of marine fungi to nitrous oxide (N_2_O) production. Fungal communities evaluated by ITS2 sequencing were dominated by the phyla *Basidiomycota* and *Ascomycota* at most depths. However, the metagenome dataset showed that about one third of the fungal community belong to early-diverging phyla. Fungal N_2_O production rates peaked at the oxic–anoxic interface of the water column, and when integrated from the oxycline to the top of the anoxic depths, fungi accounted for 18–22% of total N_2_O production. Our findings highlight the limitation of ITS-based methods typically used to investigate terrestrial fungal diversity and indicate that fungi may play an active role in marine nitrogen cycling.

## 1. Introduction

Oceanic oxygen minimum zones (OMZs) are characterized by a sharp oxycline and redox gradient in the water column [1]. As a result, OMZs support diverse microbial communities that directly impact the global biogeochemical cycling of nitrogen, carbon, sulfur, and trace metals [2,3,4,5,6]. As in many other types of marine environments, bacteria and archaea have been the focus of microbial ecology research in OMZs, whereas microbial eukaryotes, in particular fungi, have received much less attention [7,8].

An early cultivation-based survey of marine fungi in the Indian Ocean that included the Arabian Sea OMZ found *Rhodotorula rubra* and *Candida atmosphaerica* to be cosmopolitan, and the yeast population densities ranged from 0–513 cells per liter of seawater [9]. In the eastern tropical South Pacific (ETSP) OMZ off the coast of Chile, high summertime fungal biomass in the water column have been reported [10], including diatom parasites from the phylum Chytridiomycota [11]. In the third and the largest open ocean OMZ, the eastern tropical North Pacific (ETNP), protists diversity has been investigated by sequencing the V4 region of 18S small subunit rRNA genes [7], but little is known about the diversity and function of fungi in this environment. 

Fungi in the water column are generally thought to contribute to organic matter recycling, particularly in particle-associated environments [12,13,14]. High hydrolytic activity on proteinaceous substrates in large size fractions (>25 μm and >90 μm) have been reported in the water column of the ETSP and attributed to fungi given low bacterial biomass in those size fractions [15]. However, it remains unclear if fungi in the water column of OMZs play a role in nitrogen cycling. Discovered in the early 1990s, fungal denitrification is known as a process that reduces nitrate or nitrite with nitrous oxide (N_2_O) as the end-product [16,17]. This adds to the multiple other pathways and processes (ammonia oxidation, bacterial denitrification, and chemodenitrification) that can produce N_2_O [18], a potent greenhouse gas and ozone-depleting agent [19]. Many fungal strains have been found to have the ability to produce N_2_O [20], including an *Aspergillus terreus* strain isolated from the Arabian Sea OMZ [21]. In marine environments, fungal denitrification with N_2_O as the end-product has been reported from coastal marine sediment in India and Germany [22,23], but its potential contribution in the water column remains unclear.

We investigated the fungal community composition in the eastern tropical North Pacific oxygen minimum zone by sequencing the internal transcribed spacer region 2 (ITS2) and classifying shotgun metagenome reads. To estimate the fungal contribution to N_2_O production in the water column, we used a selective inhibition method combined with ^15^N-labeled tracer incubation experiments. Fungal communities evaluated by ITS2 sequencing were dominated by the phyla *Basidiomycota* and *Ascomycota* at most depths. The metagenome dataset showed that early-diverging fungi accounted for about one third of the fungal community, and the subsurface peaks of fungal abundance coincided with both cyanobacterial abundance and eukaryotic algal abundance. Incubation experiments suggest a possible role of fungi in N_2_O production in the water column of the ETNP OMZ.

## 2. Materials and Methods

### 2.1. Site Description and Seawater Filtration

In March 2018, aboard the R/V Sally Ride in the eastern tropical North Pacific oxygen minimum zone, two stations were visited to study fungal diversity and the potential fungal contribution to N_2_O production (Figure 1a). Dissolved oxygen concentration was determined using the SBE 43 dissolved oxygen sensor attached to the conductivity, temperature, and depth (CTD) rosette. Seawater was collected at multiple depths spanning from the oxycline to the anoxic depths (Figure 1b) using 30 L Niskin bottles.

To collect particulate material at different size fractions, seawater was sequentially filtered through a 47 mm Whatman Grade 541 acid-hardened cellulose filter paper (22 μm nominal particle retention rating, GE Healthcare 1541–047, Marlborough, MA, USA), a 47 mm polycarbonate filter (2.0 μm nominal pore size, Millipore Isopore TTTP-04700, Burlington, MA, USA), and a Sterivex filter (0.22 μm nominal pore size, Millipore SVGP01050, Burlington, MA, USA), using a peristaltic pump filtration at a flow rate < 50 mL/min. For each sample set, 23 to 55 L of seawater was filtered (Appendix A). Each 47 mm filter was stored in a 47 mm petri dish and flash-frozen in liquid nitrogen before storage at −80 °C. 

### 2.2. DNA Extraction

In the laboratory, DNA was extracted using the DNeasy Plant Mini Kit (QIAGEN Cat No. 69104, Germantown, MD, USA) following the DNeasy Plant Handbook [25], except that the cell disruption step was customized for our samples. Filter paper from the Sterivex filters were extracted from the plastic case using a snap-blade knife. All filters were first cut into 2 by 2 mm pieces using sterilized scissors and transferred into 2 mL screw cap tubes containing 1 mL of 0.5 mm zirconia/silica beads (Biospec products #11079105z, Bartlesville, OK, USA), 600 mL of buffer AP1, and 6 μL of RNase A. Bead beating of the samples was performed for 90 s using a Biospec Mini-BeadBeater-16 and was followed by incubation at 65 °C for 10 min. After centrifugation at 20,000× *g* for 5 min, the supernatant in each sample tube was transferred to a fresh 2 mL microcentrifuge tube and neutralized with 195 μL of Buffer P3. The remaining DNA extraction steps followed the DNeasy Plant Handbook without modifications. An extraction blank was included for each batch of extraction procedure. DNA yield was quantified using a Qubit fluorometer (ThermoFisher Scientific, Waltham, MA, USA) following the manufacturer’s instructions. All extracted DNA samples were stored at −80 °C until amplicon library construction.

### 2.3. Sequencing and Analysis of the ITS2 Region

The second region of the internal transcribed spacer (ITS2) with flanking regions in the 5.8S and 28S ribosomal RNA was targeted for amplicon library preparation following the Illumina 16S Metagenomic Sequencing Library Preparation [26] with the following modifications. The amplicon PCR was performed using primers ITS3tagmix [27] and ITS4tag001 [28] (Appendix A) with Phusion^®^ high-fidelity DNA polymerase (New England BioLabs, M0530, Ipswich, MA, USA). The thermal cycle started with 30 s at 98 °C, followed by 30 cycles of 15 s at 98 °C, 30 s at 55 °C, and 45 s at 72 °C. The final elongation at 72 °C was 10 min long. The quantity and quality of final PCR products were determined using a Qubit (ThermoFisher Scientific, Waltham, MA, USA) and TapeStation 2200 (Agilent, Santa Clara, CA, USA), respectively. Identical quantities of each sample were pooled, and the products were sequenced on an Illumina MiSeq 2 × 250 PE platform in the Biological Nanostructures Lab at UCSB. 

Raw sequence reads were first trimmed using ITSxpress to contain only the ITS2 region [29]. Trimmed reads were merged, quality filtered, dereplicated, and denoised following the USEARCH pipeline [30] to generate amplicon sequence variants (ASVs). Taxonomic assignment was performed using a combination of Naïve Bayes classifier implemented in QIIME2 [31] and BLASTn [32] against full UNITE + INSD dataset v02.02.2019 [33] and curated manually. To determine putative fungal denitrifiers in our samples, we performed a closed reference OTU picking [34] of our ASVs and ITS2 sequences from fungi tested for N_2_O production [20] using UCLUST [35] implemented in QIIME v1.9.1 [36] against the UNITE database [33]. Raw reads generated in this study are available at the National Center for Biotechnology Information (NCBI) under BioProject PRJNA623945. 

### 2.4. Analysis of Fungal Diversity and Function from Metagenomes

As an independent approach to evaluate the fungal diversity in the eastern tropical North Pacific oxygen minimum zone, we investigated a metagenome dataset sampled at a nearby station in March 2012 when the hydrographic conditions were highly similar (Appendix A) [37]. Raw reads were first filtered using the tool BBDuk Version 38.73 [38] with the options “ktrim=r ordered minlen=51 minlenfraction=0.33 mink=11 tbo tpe rcomp=f k=23 ftm=5”. Adapters were trimmed from the BBDuk-filtered reads using the tool Trimmomatic Version 0.39 [39] with the options “ILLUMINACLIP:$adapters:2:30:10 LEADING:3 TRAILING:3 SLIDINGWINDOW:4:15 MINLEN:100”. Reads that passed both quality filtering and adapter trimming were queried against the NCBI nr database using DIAMOND [40] with an e-value threshold of 1 × 10^−5^ and the options “--sensitive --min-orf 20”. The resultant NCBI taxonomy ID was used to assign taxonomy to each read. In search for the presence of the gene diagnostic for fungal N_2_O production, the cytochrome P450 nitric oxide reductase (*P450nor*) [41], we searched the metagenome assemblies under the same BioProject (PRJNA350692) for putative *P450nor* using HMMER v3.2.1 [42] against a *P450nor* profile, which includes both eukaryotic and prokaryotic cytochrome P450 genes [43]. All putative *P450nor* hits were checked against the NCBI nr database using blastp [32] to determine taxonomy. 

### 2.5. Measurements of Potential N_2_O Production Rates

To measure potential rates of N_2_O production, parallel incubation experiments were performed by adding 0.1 mL of 5 mM 99% pure ^15^N-labeled potassium nitrate or 0.1 mL of 0.8 mM 99% pure ^15^N-labeled ammonium chloride (Cambridge Isotopes, Cambridge, MA, USA) to 120 mL glass serum bottles containing freshly collected seawater. To minimize the introduction of atmospheric oxygen, each bottle was overflown three times its volume with water directly from Niskin bottles before filling and crimp-sealing it with grey butyl rubber stopper and aluminum caps. Headspace was created in each bottle by replacing 2 mL of seawater with ultra-high pure helium (Airgas HE UHP300, Radnor, PA, USA). End points were taken by adding 1 mL of 50% (*w*/*v*) zinc chloride to separate parallel incubations approximately 0 and 24 h after incubations began at Station 1 and 0, 12, and 24 h after incubations began at Station 2. The concentration of ammonium in seawater was measured onboard according to the fluorometric method of Holmes et al. [44], with a detection limit of 15 nmol L^−1^. Nitrate concentrations were assayed using a Lachat Flow Injection Analyzer at the Analytical lab at the Marine Science Institute, University of California, Santa Barbara following standard analytical methods [45]. The detection limit of nitrate was 0.2 μmol L^−1^.

The potential contribution of fungal N_2_O production to total N_2_O production was determined by incubations with ^15^N-labeled nitrate (^15^NO_3_^−^) and chloramphenicol (87.7 mg L^−1^ final concentration), which was applied to each incubation one hour before the addition of ^15^N tracers to inhibit all prokaryotic activities. All solutions added to the incubation bottles were purged by ultra-high pure helium for one hour at a flow rate of 40 mL min^−1^. The difference between fungal N_2_O production and total N_2_O production from incubations with ^15^NO_3_^−^ is attributed to bacterial denitrification (Appendix A). N_2_O production rates measured in incubations with ^15^N-labeled ammonium (^15^NH_4_^+^) are attributed to archaea and/or bacterial nitrification. 

The quantity and isotopic composition of dissolved N_2_O was determined using a Delta XP isotope ratio mass spectrometer coupled to a purge-and-trap front end. The detection limit was 1.0 nmol N, and the precision for δ^15^N was 2.0‰ (*n* ≥ 3). The rate of N_2_O production (R_N2O_) was calculated from the equation [46]: RN2O=dN152O/dtf15×V
where d^15^N_2_O/dt is the rate of ^15^N_2_O production determined from linear regression of the amount of ^15^N_2_O against time, f^15^ is the fraction of ^15^N labeled substrate, and V is the volume of the incubation. The amount of ^15^N_2_O at each time point is calculated from the equation: N152O=N2O×(δN15_N2O1000+1)×Rref1+(δN15_N2O1000+1)×Rref
where N_2_O is the amount of nitrous oxide determined from in-house N_2_O concentration standards (Appendix A), δ^15^N_N_2_O is the bulk isotopic composition of sample N_2_O, and R_ref_ is isotopic composition of reference gas. The linearity effect for the range of N_2_O measured was negligible compared to the enriched δ^15^N_N_2_O measured from our incubation samples (Appendix A). 

## 3. Results

### 3.1. Fungal Diversity Assessed by Sequencing the ITS2 Region

Assessment of the fungal community in the eastern tropical North Pacific using the internal transcribed spacer region 2 (ITS2) revealed that taxa from the phyla *Basidiomycota* and *Ascomycota* dominated at most depths at both stations (Figure 2). The relative abundance of *Basidiomycota* was higher than *Ascomycota* at nearly all depths (Appendix A). The most prevalent and abundant taxon is the basidiomycetous yeast family Sporidiobolaceae, primarily consisting of the genera *Rhodotorula*, *Rhodosporidiobolus*, and *Sporobolomyces* (Appendix A). Sporidiobolaceae tend to have a higher relative abundance in the 0.2–2 μm size fraction. On the other hand, *Aureobasidium* (Ascomycota) and Exobasidiomycetes (Basidiomycota, primarily *Meira*) were enriched in the larger size fractions (2–22 and >22 μm). In contrast, the basidiomycetous yeast *Malassezia*, when detected, were enriched only in the 2–22 μm size fraction. At Station 2, most of the fungal community cannot be classified even at the phylum level based on ITS2 sequences, indicating the presence of novel fungal lineages in the oxycline of ETNP oxygen minimum zone.

### 3.2. Fungal Diversity Assessed from Metagenomes

Fungal community composition assessed by metagenomic reads also showed the dominance of Dikarya fungi, but in contrast to the results from ITS2 sequencing, the relative abundance of Ascomycota was consistently higher than that of Basidiomycota (Figure 3). Surprisingly, over one third of the fungal community belong to early-diverging phyla including Mucoromycota, Zoopagomycota, Chytridiomycota, Blastocladiomycota, Cryptomycota, and Microsporidia. The fungal community composition from 60 to 300 m was uniform, while there was a trend of increasing relative abundance for Ascomycota with depth. 

### 3.3. Relative Abundance of Fungi Compared to Other Taxa

Mining the metagenomes allowed us to estimate the relative abundance of fungi and other taxa as part of the overall microbial community. In each metagenome, 23–57% of the reads had a positive hit against the NCBI nr database with an e-value threshold of 1 × 10^−5^ (Appendix A). Fungi accounted for 0.02–0.22% of all classifiable reads, with a subsurface peak at 70 m (Figure 4a). The relative abundance of fungal reads decreased with depth below 70 m but showed a small increase at 140 m, which was the top of the oxygen deficient layer of the water column. Ciliophora, Oomycetes, and Dinophyceae were similar to fungi in both abundance and distribution. The abundance of reads classified as eukaryotic algae also showed a subsurface peak at 70 m (Figure 4b). The abundance of cyanobacteria decreased with depth overall, but there was an increase at 110 m, coinciding with the deep chlorophyll maximum situated immediately above the anoxic depths. 

### 3.4. Potential Contribution of Fungal N_2_O Production

N_2_O production rates from incubation experiments with either ^15^NO_3_^−^ or ^15^NH_4_^+^ were detected at multiple oxycline depths of the ETNP oxygen minimum zone (OMZ), and the maximum rate was found at the oxic–anoxic interface (Figure 5). The rates of N_2_O production from incubations with ^15^NO_3_^−^ and chloramphenicol were used as an approximation of fungal N_2_O production, and they ranged from 0% of total N_2_O production from ^15^NO_3_^−^ at 275 m at Station 2 to 56% of total N_2_O production from ^15^NO_3_^−^ at 90 m at Station 1. N_2_O production from incubations with ^15^NO_3_^−^, both with and without chloramphenicol, was lower at elevated in situ oxygen (O_2_) concentration. When integrated from the oxycline (60 m at Station 1 and 90 m at Station 2) to the oxic–anoxic interface, fungal N_2_O production approximated by incubations with ^15^NO_3_^−^, and chloramphenicol accounted for 18–22% of total N_2_O production (Appendix A).

### 3.5. Search for Fungal Denitrifiers and Functional Genes

To search for putative fungal denitrifiers in the water column of the eastern tropical North Pacific OMZ, we first analyzed the ITS2 amplicon dataset generated in this study. Of all 237 ASVs, only one (ASV211) was clustered at 97% similarity level with the ITS2 sequences from fungal strains previously shown to produce N_2_O (*Chaetomium* sp.) [20]; this ASV was present only in the 0.2–2 μm size fraction at 83 m from Station 1 and at a relative abundance of 0.2% (Appendix A). However, one ASV (ASV8) was 96.95% similar to the ITS2 region of the N_2_O-producing *Penicillium melinii* [20], which was isolated from seawater [48]. The relative abundance of ASV8 ranged from 0.1 to 8.5% at hypoxic and anoxic depths (Appendix A), and it belongs to the same family (Aspergillaceae) as the N_2_O-producing fungal strain isolated from the Arabian Sea OMZ [21]. While this putative denitrifier could potentially produce N_2_O in the water column of the OMZ, its potential contribution to the total N_2_O production is estimated to be less than 0.3% given the low N_2_O yield by *P. melinii* in the laboratory (Appendix B). Continuing the search for putative fungal denitrifiers, we analyzed a previously published March 2012 metagenome dataset from a nearby station [37] for the occurrence of *P450nor*. All *P450nor* hits identified by a hidden Markov model (HMM) profile search [43] were prokaryotic genes, most of which from the phylum Actinobacteria (Appendix A).

## 4. Discussion

### 4.1. Fungal Diversity in Different Size Fractions

In the eastern Tropical North Pacific (ETNP) oxygen minimum zone, the most prevalent and abundant fungal taxa in our ITS2-based survey was the basidiomycetous yeast from the family Sporidiobolaceae, which are usually characterized by the production of carotenoids that color their cells red, pink, or orange and, hence, have the name “red yeasts”. The high relative abundance of red yeasts in the 0.2–2 μm size fraction (Figure 2) is consistent with their typically small cell size [49]. A previous study on yeasts in the Indian Ocean, which includes the Arabian Sea oxygen minimum zone in the northern part, has also found red yeasts to be the predominant taxa [9]. On the other hand, *Aureobasidium* (Ascomycota) and Exobasidiomycetes (Basidiomycota, primarily *Meira*) were enriched in the larger size fractions (2–22 µm and >22 μm), consistent with a larger size [50,51], and indicating a likely association with particles. In contrast, the basidiomycetous yeast *Malassezia*, when detected, were enriched only in the 2–22 μm size fraction, suggesting that they were not associated with particles larger than 22 μm.

There is increasing recognition of marine fungi as a key component of the marine microbiome and biogeochemical cycles [12,13,15,52,53], but our knowledge about their diversity and function is far less compared to other microbial eukaryotes [54]. This is particularly the case in the water column of the open ocean, where metabarcoding surveys of fungal diversity are often unable to classify most of the fungal community [55,56]. In one of the two stations sampled in this study, we also could not resolve the taxonomy for most of the fungal community (Figure 2), highlighting the limitation of an ITS-based approach to study fungal diversity. Because the public databases for ITS sequences are primarily based on studies of terrestrial environments or fungal strains, this suggests that the open ocean water column harbors previously undiscovered lineages of fungi. 

### 4.2. Ecology of Marine Fungi in the Oxygen Minimum Zone

By classifying metagenome reads, we showed that about a third of the fungal community were from early-diverging lineages [57], including Mucoromycota, Zoopagomycota, Chytridiomycota, Blastocladiomycota, Cryptomycota, and Microsporidia (Figure 3). Such diversity was previously undiscovered due to both the difficulty in cultivation and the consequent lack of representation in ITS databases. It is unclear what ecological roles these early-diverging fungi play, except for Chytridiomycota, which are typically associated with phytoplankton such as diatoms [58]. Nevertheless, the depth profile of the relative abundance of fungi in relation to other microbial taxa provides a hint (Figure 4). The cooccurrence of subsurface peaks (at 70 m) of the relative abundance of fungi and eukaryotic algae suggests that fungi are directly associated with eukaryotic algae, perhaps as parasites. A secondary peak of the relative abundance of fungi (at 140 m) was observed below that of the deep chlorophyll maximum (at 110 m) typically found at the oxic–anoxic interface of oxygen deficient waters [59]. Hence, the fungal communities at the secondary peak (at 140 m) are likely predominantly saprotrophic, feeding on the particles formed at the base of the mixed layer. 

Surprisingly, we did not observe a pronounced effect of oxygen concentration on the fungal community composition, evaluated by either ITS2 amplicon sequencing or metagenomes. In contrast, the protist community in the anoxic depth of the ETNP and eastern tropical South Pacific oxygen minimum zone was enriched in Syndiniales, euglenozoan flagellates, and acantharean radiolarians [7,8]. This may be a result of the versatile respiratory/fermentative metabolisms fungi possess, but it could also be due to the inability of the methods used in this study to detect novel fungal lineages from the oxygen deficient waters. The metagenome-based approach avoids the typical biases associated with amplicon sequencing such as primer bias, but it is limited by the sequences available in the chosen database (NCBI nt in this study).

### 4.3. Fungal N_2_O Production in the Oxygen Minimum Zone

In order to distinguish the N_2_O production by fungi from bacteria and archaea, we combined ^15^N tracer incubation experiments with selective inhibition using the antibiotic chloramphenicol (Appendix A). Chloramphenicol is a broad-spectrum antibiotic that has been used to isolate anaerobic gut fungi from the rumen microbiome [60], and its final concentration used in our incubations was scaled by the typical bacterial cell density in the rumen vs. seawater. Nonetheless, we did not make direct measurements of the specificity and effectiveness of chloramphenicol on inhibiting bacterial and archaeal activities in seawater. Therefore, it is possible that the N_2_O production rates we measured from incubations with ^15^NO_3_^−^ and chloramphenicol include N_2_O from partially inhibited bacterial denitrification. Consequently, we conservatively interpret those rates as an upper limit of potential fungal N_2_O production.

When integrated from the oxycline to the oxic–anoxic interface, fungal N_2_O production accounted for 18–22% of total N_2_O production (Appendix A). While we interpret this as the maximum possible contribution of fungi to N_2_O production, we suggest that fungal denitrification could be an important pathway for N_2_O production in oceanic oxygen minimum zones. Denitrification, the sequential reduction in NO_3_^−^ to N_2_, is known to be inhibited by trace amounts of O_2_ [61]. In OMZs, it was demonstrated that 297 nM of O_2_ repressed 50% of total N_2_O production at the oxic–anoxic interface [62]. In this study, N_2_O produced in incubation with ^15^NO_3_^−^ (primarily via denitrification) was inhibited by increasing levels of in situ O_2_ concentration. However, the inhibitory effect of O_2_ appeared to be less pronounced on fungal denitrification than on bacterial denitrification at low levels, revealing a potential niche (0.0 < O_2_ < 0.93 μM) for fungi capable of denitrification. Since this potential niche is shallower than the oxygen deficient waters, the potential fungal N_2_O production can have a higher chance of reaching the ocean–atmosphere interface than N_2_O produced at deeper depths. Alternatively, N_2_O produced by fungi can be taken up by bacteria with the atypical (“Clade II”) nitrous oxide reductase [63] (primarily Flavobacteria and Chloroflexi), which was shown to have a peak in relative abundance at depths immediately above the oxic–anoxic interface [37].

### 4.4. Molecular Evidence for Fungal N_2_O Production

The search for putative fungal denitrifiers using ITS2 sequence identity suggests an extremely low abundance of known N_2_O-producing fungi in the water column of oxygen minimum zones. However, it should be noted that the collection of N_2_O-producing fungi used to identify these putative denitrifiers consists of soil fungi exclusively [20], so fungal lineages capable of N_2_O production from the open ocean were likely excluded. Therefore, there may be other N_2_O-producing fungi from the ETNP OMZ unidentified by this approach.

The absence of fungal *P450nor* in the metagenomic dataset we queried may be attributed to insufficient sequencing depth, given the low percentage (0.02–0.22%) of fungal reads classified by DIAMOND against the NCBI nr database [40] (Figure 4a). Even under the most simplistic assumption that there was only one fungal species present and it possessed *P450nor* in its genome, the sequencing depth in most samples was insufficient to recover just one copy of fungal *P450nor* (Appendix A). Additionally, most DNA extraction protocols applied in published metagenome studies (including [37]) are not customized for disrupting chitinous cell walls. This likely resulted in under-sampling DNA from fungi, of which the biomass is low in the ocean water column (0.01–0.12 μg of carbon L^−1^) [10], especially compared to bacteria (estimated average of 10.5 μg of carbon L^−1^) [64]. Finally, it should be noted that the detection of fungal *P450nor* genes in metagenomes does not necessarily imply fungal denitrification, as *P450nor* in certain fungal genomes appear to be involved in secondary metabolisms instead [43].

## 5. Conclusions

Our findings highlight the previously unrecognized fungal diversity in the eastern tropical North Pacific oxygen minimum zone, particularly from the early-diverging taxa as revealed by analysis of shotgun metagenomes. The depth distribution pattern of fungi in relation to cyanobacteria and eukaryotic algae suggest direct association in the mixed layer of the water column and indirect feeding below the deep chlorophyll maximum. Given the limitations of the selective inhibition method using chloramphenicol, we estimate that fungi contribute no more than 18–22% to total N_2_O production from the oxycline to the oxic–anoxic interface. It remains challenging for omics-based approaches to provide molecular evidence for fungal denitrification, partially because current databases and recent studies have primarily focused on terrestrial fungi. Overcoming the shortfalls of existing methods necessitates approaches that can target fungal diversity and function, such as the use of RNA-seq combined with eukaryotic messenger RNA enrichment or the combination of fluorescence-activated cell sorting and (meta)genome sequencing. These new methods can increase the likelihood of capturing genetic evidence for fungal activities and functional diversity.

## Figures and Tables

**Figure 1 jof-07-00218-f001:**
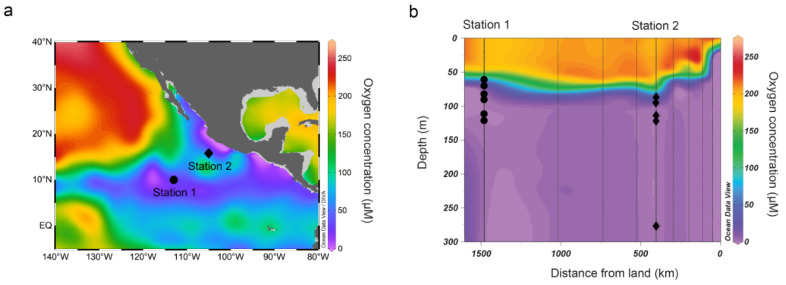
(**a**) Sampling locations in the eastern tropical North Pacific (ETNP) oxygen minimum zone. Color contour shows oxygen concentrations at 100 m depth from World Ocean Atlas 2013 (March average from 1955–2012) [24]. (**b**) Depths sampled for nitrous oxide (N_2_O) production experiments are marked by filled symbols. Color contour shows oxygen concentrations measured during this cruise using a Seabird SBE 43 dissolved oxygen sensor.

**Figure 2 jof-07-00218-f002:**
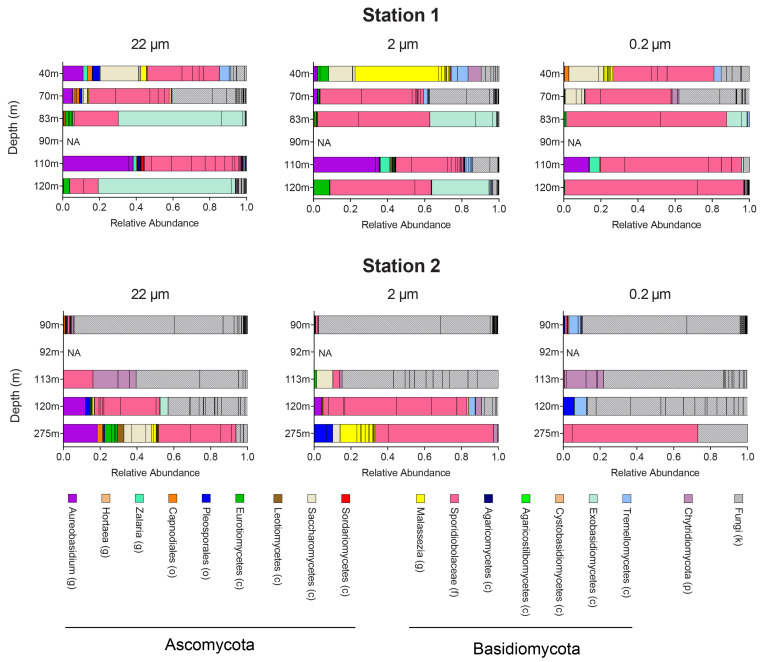
Fungal community composition for three different size fractions including >22 μm (left), 2–22 μm (middle), and 0.2–2 μm (right) at Stations 1 and 2. Each bar represents an amplicon sequence variant (ASV) of the internal transcribed spacer region 2 (ITS2). NA: not available. The level of each taxonomic assignment is indicated in the parenthesis behind the names: g for genus, f for family, o for order, c for class, p for phylum, and k for kingdom.

**Figure 3 jof-07-00218-f003:**
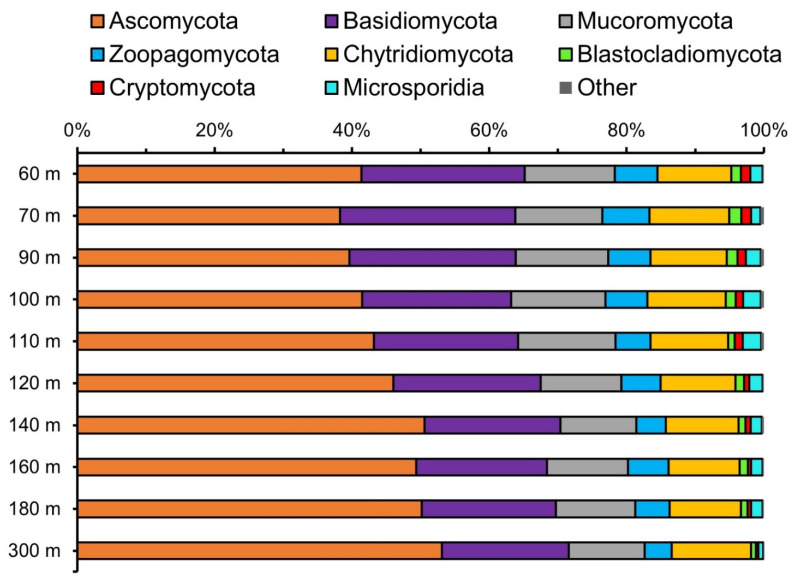
The relative abundance of each fungal phylum identified in the metagenomes [37] collected from a location near Station 2 in this study.

**Figure 4 jof-07-00218-f004:**
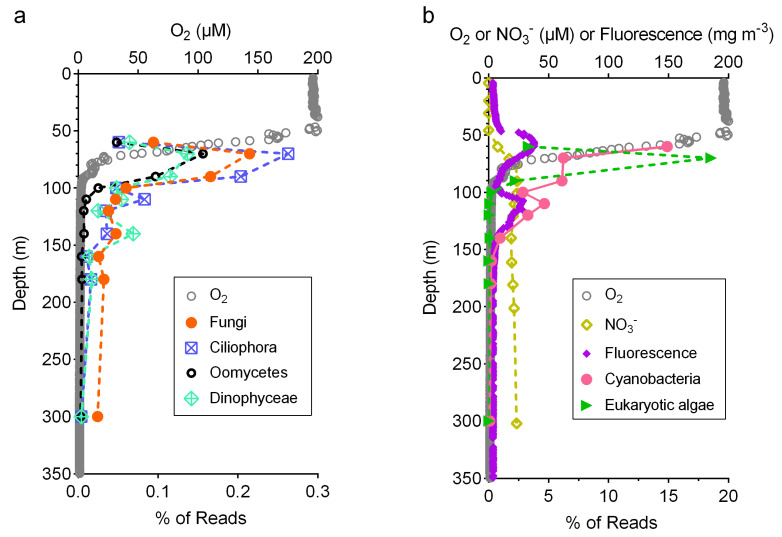
(**a**) Dissolved oxygen concentration (μM) and the relative abundance of reads classified as Fungi, Ciliophora, Oomycetes, and Dinophyceae from the metagenomes [37] collected from a location near Station 2 in this study; (**b**) dissolved oxygen concentration (μM), nitrate (NO_3_^−^) concentration (μM), fluorescence (mg m^−3^) [47], and the relative abundance of reads classified as Cyanobacteria and eukaryotic algae. Eukaryotic algae include Haptophyta, Bacillariophyta, Rhodophyta, Cryptophyta, and Viridiplantae.

**Figure 5 jof-07-00218-f005:**
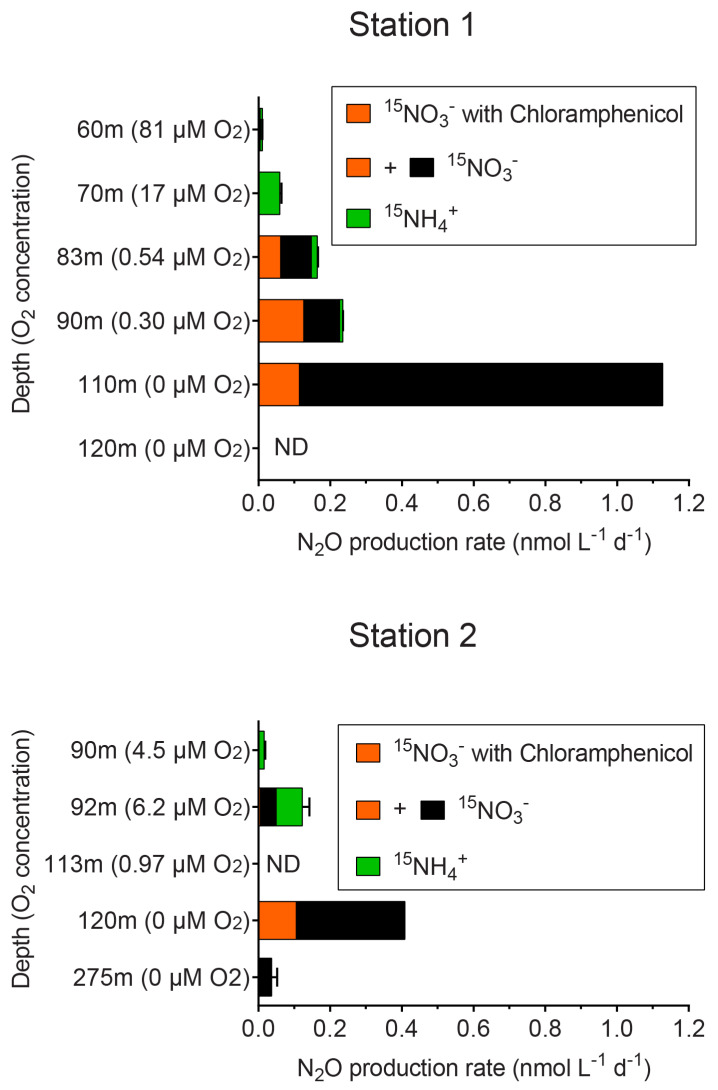
N_2_O production rates measured from incubation experiments with ^15^NO_3_^−^ only (orange and black bars combined), with ^15^NO_3_^−^ and chloramphenicol (orange bars), and with ^15^NH_4_^+^ (green bars). Chloramphenicol was intended to inhibit bacterial and archaeal activity.

## Data Availability

Raw reads generated in this study are available at NCBI under BioProject PRJNA623945.

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
