# Peer review of "Diversity and N2O Production Potential of Fungi in an Oceanic Oxygen Minimum Zone"

_jof, 2021, doi:10.3390/jof7030218_

Round 1

Reviewer 1 Report

The manuscript by Peng et. al. describes fungal diversity in a marine anoxic zone. Overall, fungal diversity has been neglected in AMZ research, so evaluating the 'who' and 'where' is of interest. The range of samples is somewhat limited, with a focus on the ETNP. This is somewhat understandable. Fungal populations might be excluded from previous collection of samples using pre-filtration. Overall I don't really have to many qualms with the manuscript. I think it would make a nice addition.

Author Response

We thank the reviewer's comments on our manuscript.

Reviewer 2 Report

The paper submitted by Peng and Valentine at the Journal of Fungi, investigates the fungal community composition in the ENTP oxygen minimum zone (OMZ), addresses the role of fungal denitrification in this particular OMZ and finally estimates the fungal contribution to the N2O pools between 60 to ~300 m water column depths using isotopically labeled 15N-NO3 and 15N-NH4. The role of fungal denitrification in the marine environment (from coastal and hydrothermal sediments up to water columns) is generally poorly understood. Studies like that of Peng and Valentine could provide insights on how fungi can shape the fate of different nitrogen pools in the marine environments, and also amend our knowledge on marine fungal diversity. This reviewer is pleased with the quality and the results of the work presented by Peng and Valentine. 

There are some points/suggestions that this reviewer would like to address to the authors:

a) Due to the truncated fungal denitrification pathway it would be interesting to see what bacteria co-exist at the depths were the authors report the higher rates of N2O production. This reviewer believes that fungal-bacterial consortia is possible to exist in OMZs and shape the N2 production rates, via the N2O produced by fungi and potentially uptaken by bacterial denitrifiers. The ideal would be if the authors had information on bacterial 16S rDNA for their sampled sites (either free-living or particle associated), or if they could comment from previously published literature at the sites where the authors conducted their experiments on the bacterial communities. Regardless, a statement on synergistic denitrification in the OMZs either in the discussion or the conclusions, might be beneficial for the reader.

b) This reviewer is aware how difficult and constrained can be fungal taxonomic annotation. However, did the authors tried to do taxonomic assignment of their ASVs using BLAST against the UNITE classifier? This might give a better resolution on the unidentified fungi that dominate especially at Station 2.

c) The reader might also benefit if the authors provide the in situ NO3 concentrations, if available, and similarly to Figure 4a if they authors could provide relative abundance vs. NO3/NH4 (if applicable and informative).

d) Finally this reviewer agrees that absence of P450nor can be due to low sequencing depth and DNA undersampling for fungi (primarily from low biomass rather than beat beading and cell lysis that was performed in this study-with the caveat that if sonication was applied it is possible the authors might have better fungal DNA yield; Kumar& Mungunthan, 2018), however even if P450nor was detected it can be used only as a proxy of dentrification, since the activity of the enzyme can be compromised from the available carbon source under denitrification (Watsuji et al., 2003).  

Author Response

We thank the reviewer for the comments on our manuscript. Below are responses to the specific points the reviewer raised.

a) Due to the truncated fungal denitrification pathway it would be interesting to see what bacteria co-exist at the depths were the authors report the higher rates of N2O production. This reviewer believes that fungal-bacterial consortia is possible to exist in OMZs and shape the N2 production rates, via the N2O produced by fungi and potentially uptaken by bacterial denitrifiers. The ideal would be if the authors had information on bacterial 16S rDNA for their sampled sites (either free-living or particle associated), or if they could comment from previously published literature at the sites where the authors conducted their experiments on the bacterial communities. Regardless, a statement on synergistic denitrification in the OMZs either in the discussion or the conclusions, might be beneficial for the reader.

Thanks for the suggestion. We agree that it is possible that the N2O produced by fungi can be taken up by bacteria. A previous study at the same site (Fuchsman et al. 2017) investigated the nitrous oxide reductase community using metagenomics, which showed that Flavobacteria and Chloroflexi (with atypical nitrous oxide reductase subunit Z) peaked in relative abundance at depths immediately above the oxic-anoxic interface, supporting this synergistic denitrification possibility. We have included in the discussion a statement on synergistic denitrification in OMZs (lines 388-391).

b) This reviewer is aware how difficult and constrained can be fungal taxonomic annotation. However, did the authors tried to do taxonomic assignment of their ASVs using BLAST against the UNITE classifier? This might give a better resolution on the unidentified fungi that dominate especially at Station 2.

We appreciate the suggestion. Yes, we have tried BLASTing our ASVs against the latest version of the UNITE database. The most abundant ASVs at Station 2 can only be classified to the level of the fungal kingdom.

c) The reader might also benefit if the authors provide the in situ NO3 concentrations, if available, and similarly to Figure 4a if they authors could provide relative abundance vs. NO3/NH4 (if applicable and informative).

Thanks for the suggestions, the in situ NO3- concentrations is now included in Figure 4b. NH4+ concentrations were below detection at all depths so not included.

d) Finally this reviewer agrees that absence of P450nor can be due to low sequencing depth and DNA undersampling for fungi (primarily from low biomass rather than beat beading and cell lysis that was performed in this study-with the caveat that if sonication was applied it is possible the authors might have better fungal DNA yield; Kumar& Mungunthan, 2018), however even if P450nor was detected it can be used only as a proxy of dentrification, since the activity of the enzyme can be compromised from the available carbon source under denitrification (Watsuji et al., 2003).  

Thanks for the suggestion. We would like to clarify that cell lysis by beat beating was performed in this study to extract DNA for ITS sequencing, but the metagenomes we investigated from Fuchsman et al. (2017) for P450nor was sequenced using DNA extracted without bead beating. Therefore, it is possible that the metagenome libraries we investigated were biased against hard-to-lyse fungal cells. We agree that the detection of P450nor may not be an accurate proxy for P450nor and have edited the corresponding section in the discussion (lines 412-414).